# Borehole diameter shrinkage rule considering rheological properties and its effect on gas extraction

Fuchang Hao[1,2]*, Lijuan Sun[1], Fajun Zhao[1,2]

1 State Key Laboratory Cultivation Base for Gas Geology and Gas Control, Henan Polytechnic University, Jiaozuo, Henan, China, 2 The Collaborative Innovation Center of Coal Safety Production of Henan Province State Key, Jiaozuo, Henan, China

* haofuchang@163.com

## Abstract

To study the shrinkage rule of borehole diameter and its effect on gas extraction, a visco-elastoplastic model for boreholes considering strain softening and the dilatancy characteristic was established to obtain the expressions of the coal stress, variation in diameter, and pressure relief range. The stress distribution and pressure relief effect of the boreholes in soft and hard coal seams were comparatively analyzed. The shrinkage rule of the borehole diameter was studied. The reasons for the rapid reduction in the extraction concentration of the borehole in soft coal seams were described. A technology of improving the gas extraction effect in soft coal seams was developed. The research results showed that the radius of the plastic softening zone is 0.405 m for a borehole in a soft coal seam and 0.224 m for that in a hard coal seam. This indicates that the borehole in a soft coal seam has a better pressure relief effect. The boreholes in both hard and soft coal seams will incur a shrinkage phenomenon; however, the soft coal seam has low strength and a weak ability to resist damage, and thus the surrounding coal will have a more intense creep deformation, leading to an instability failure during a short period of time and thus, a blocking of the extraction channel, thereby causing a rapid reduction in the gas extraction concentration. The borehole in a hard coal seam also shows a shrinkage phenomenon, but remains in a stable state without a blockage; thus, high-concentration gas can be extracted from this borehole for a long period of time. The geo-stress and coal strength are the two main factors controlling the amplitude of borehole shrinkage. From an increase in stress, the borehole in a hard coal seam shows a more intense creep deformation in a deep mine, which may lead to blockage. The key to improving the gas extraction effect in soft coal seams is to maintain a smooth extraction channel. The full screen pipe is installed through the drill pipe to retain an extraction channel, leading to an average gas extraction increase from 0.043 m³/min to 0.12 m³/min, an increase of 2.77 times. These research results are consistent with actual production, and can provide theoretical guidance for determining the gas extraction parameters.

**Data Availability Statement:** All relevant data are within the manuscript and its Supporting Information files.

**Funding:** This work was funded by the National Natural Science Funds of China (No. 51604092),

the Henan Provincial Natural Science Founs (No. 182300410128), and the Henan Province Science and Technology Project (No. 182102310828), and the Fundamental Research Funds for the Universities of Henan Province.

## 1. Introduction

Gas extraction is the most important prevention and control measure of coal and gas outburst [1–4]. On the one hand, the role of a borehole can be to relieve the pressure of the coal and reduce the power of coal and gas outburst in releasing the elastic potential. On the other hand, it can provide a gas extraction channel to extract the gas and reduce the gas pressure and gas content of the coal seam, releasing the gas potential. It is generally believed that the main factors affecting the gas extraction effect are coal seam permeability and borehole sealing quality, but the practice of gas extraction has shown that the extraction channel is another important factor [5–8]. A coal body can be divided into two types according to its strength: coal whose uniaxial compressive strength $\sigma_c$ is lower than 5.0MPa is defined as soft coal, whereas coal with $\sigma_c$ of greater than 5.0MPa is defined as hard coal. The practice of gas extraction has shown that its effect varies significantly between the soft and the hard coal seams, a high concentration of gas can be extracted from the boreholes in a hard coal seam for a long period of time, and the gas concentration can be maintained at over 20% three months after extraction. However, the concentration of gas extracted from a borehole in a soft coal seam decreases quickly, and may reduce to 5% after only 10 days, after which, even if the extraction time is extended, it will be difficult to improve the gas extraction effect. An analysis of the reasons behind this effect shows that, owing to the rheological properties of coal, the surrounding coal will incur creep deformation, and the diameter of the borehole will experience shrinkage over time. Creep deformation is more severe for a soft coal seam owing to its low strength and weak ability to resist damage, possibly leading to an instability failure during a short period of time, thereby blocking the gas extraction channel and decreasing the extraction concentration [9–11]. A hard coal seam has a strong capability to resist destruction, and thus a borehole in such a seam can remain in a stable state despite a certain amount of shrinkage, and a high concentration of gas can be extracted from the borehole during a long period of time. Therefore, studies on the rule of borehole diameter shrinkage are of significance when optimizing the borehole layout and improving the gas extraction effect.

Rock creep deformation has a strong influence on the development of gas extraction, coalbed methane, shale gas, and tight oil [12, 13]. A number of engineering studies have been conducted on rock creep deformation, including theoretical, experimental, and numerical methods [14–21]. A borehole stability analysis mainly uses the linear elastic-brittle or porous elastic theory, and when the stress around a borehole violates a particular rock failure criterion, it is assumed that the borehole has collapsed [22–25]. Based on further research on an analytical method of the creep deformation of circular holes, the rheological constitutive model of surrounding rock underwent the development from a less-viscous viscoelastic model to a more rational visco-elastoplastic model [26–29]. The results show that coal is a type of elastoplastic body with low strength, and has strain softening and dilatation characteristics. A viscoelastoplastic model for boreholes considering strain softening and the dilatancy characteristic has been little studied. Therefore, it is very important to establish a visco-elastoplastic model for boreholes considering considering strain softening and the dilatancy characteristic, and to study the borehole diameter shrinkage rule.

## 2. Theoretical analysis

### 2.1. Mechanical model

The stress distribution around boreholes is shown in Fig 1. To simplify the model, the following assumptions are given: (1) The borehole is in a hydrostatic pressure field with an initial stress of $\sigma_0$, (2) the coal is homogeneous and isotropic, (3) the borehole is infinitely long, and is

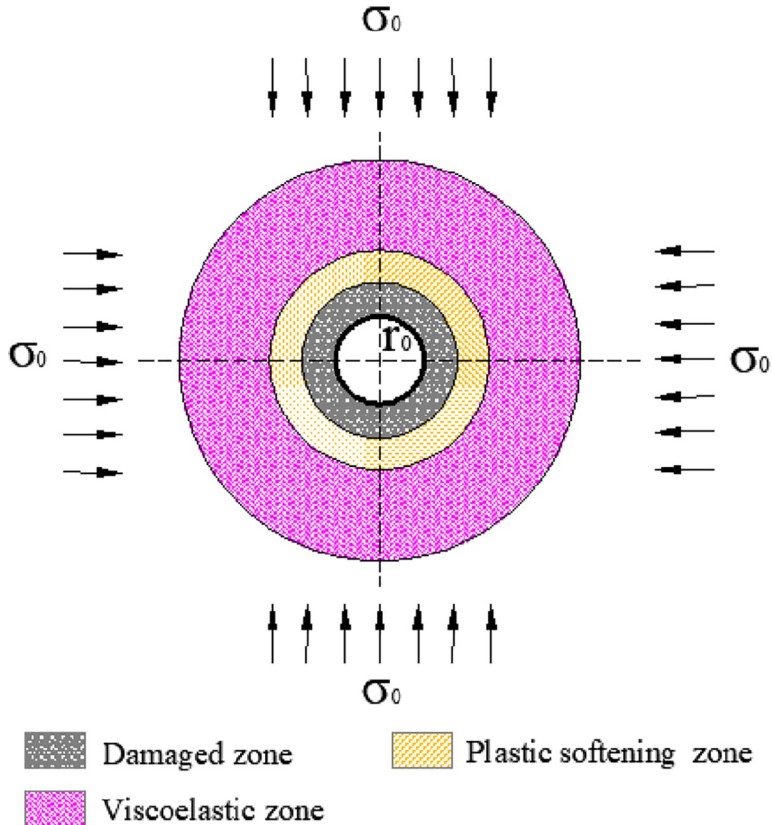

**Fig 1.** Mechanical model of surrounding coal.

processed as the plane strain problem, and (4) the surrounding coal is divided into the damaged zone, plastic softening zone, and viscoelastic zone, which are expressed as $b$, $p$, and $e$, respectively, through which the plastic softening zone incurs strain softening, and the plastic softening zone and the damaged zone have a dilatancy characteristic. Considering that the surrounding coal first incurs viscoelastic deformation over time and then enters into plastic deformation, visco-elastoplastic softening models were established. The Poynting-Thomson model was adopted for the viscoelastic zone, and the plastic flow model was applied to the plastic softening zone and the damaged zone.

## 2.2. Constitutive relation and yield criterion

**(1) Basic equations under polar coordinate system.** The equilibrium equation is

$$\frac{d\sigma_r}{dr} + \frac{\sigma_r - \sigma_\theta}{r} = 0, \tag{1}$$

and the geometric equation is

$$\varepsilon_r = \frac{du}{dr}, \ \varepsilon_\theta = \frac{u}{r}, \tag{2}$$

where $\sigma_r$ and $\sigma_\theta$ are the radial and tangential stresses, respectively; $\varepsilon_r$ and $\varepsilon_\theta$ are the radial and tangential strains; and $u$ is the radial displacement.

**(2) Yield criterion.** For coal deformation in the viscoelastic zone, the Poynting-Thomson model is employed.

For the coal yield in the plastic softening zone, the Mohr-Coulomb strength criterion is employed, namely,

$$\sigma_\theta^p = K_p \sigma_r^p + \sigma_c^p, \tag{3}$$

where $K_p = (1+\sin\phi)/(1-\sin\phi)$; $\phi$ is the internal friction angle of the coal; $\sigma_r^p$ and $\sigma_\theta^p$ are the radial and tangential stresses of the coal in the plastic softening zone, respectively; and $\sigma_c^p$ is the compressive strength of coal in the plastic softening zone.

In the plastic softening zone, the compressive strength $\sigma_c^p$ is a function of strain. The greater the deformation of the coal is, the lower $\sigma_c^p$ is. The stress-strain three-line segment model (Fig 2) is employed to express the relationship between $\sigma_c^p$ and the strain.

$$\sigma_c^p = \sigma_c - M_c[\varepsilon_\theta - (\varepsilon_\theta^e)_{r=R_p(t)}], \tag{4}$$

where $M_c$ is the softening modulus; $R_p(t)$ is the radius of the plastic softening zone, and is a function of time; and $\varepsilon_\theta^e$ is the tangential strain in the viscoelastic zone.

In the damaged zone, the residual strength $\sigma_c^*$ is considered to be fixed, and employs the Mohr-Coulomb strength criterion, namely,

$$\sigma_\theta^b = K_p \sigma_r^b + \sigma_c^* \tag{5}$$

where $\sigma_\theta^b$ and $\sigma_r^b$ are the radial and tangential stresses in the damaged zone, respectively; and $\sigma_c^*$ is the residual strength of the coal.

**(3) Dilatancy characteristic.** The results of a uniaxial compression test of coal show that the volume of coal in the pre-peak zone incurs little change, and its dilatancy mainly occurs in the post peak zone. For the surrounding coal, the volume deformation of coal in the viscoelastic zone is negligible, and the dilatancy characteristic exists in both the plastic softening zone and the damaged zone. For the plastic softening zone,

$$\Delta\varepsilon_r^p + \eta_1 \Delta\varepsilon_\theta^p = 0, \tag{6}$$

where $\Delta\varepsilon_r^p$ and $\Delta\varepsilon_\theta^p$ are the radial and tangential strain increments of coal in the plastic softening zone, respectively; $\eta_1$ is the dilatancy coefficient in the plastic softening zone, and $\eta_1 = (1+\sin\varphi)/(1-\sin\varphi)$, $\phi$ is the expansion angle of coal。

In the damaged zone,

$$\Delta\varepsilon_r^b + \eta_2 \Delta\varepsilon_\theta^b = 0, \tag{7}$$

where $\Delta\varepsilon_r^b$ and $\Delta\varepsilon_\theta^b$ are the radial and tangential strain increments of coal in the damaged zone, respectively; and $\eta_2$ is the dilatancy coefficient of coal in the damaged zone.

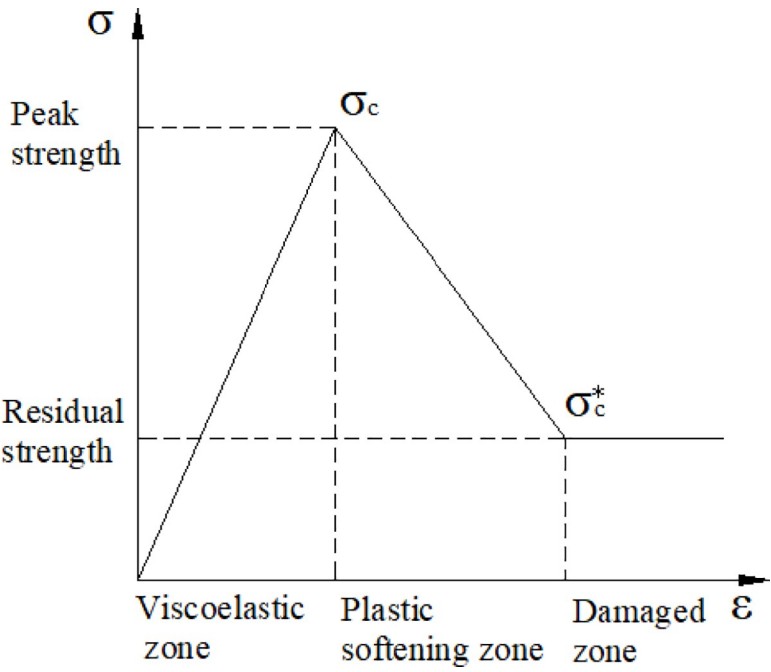

**Fig 2.** Three-line segment softening model.

## 3. Viscoelastic plastic analysis of surrounding coal

### 3.1. Stress analysis of surrounding coal

**(1) Viscoelastic zone.** For the viscoelastic zone, the Poynting–Thomson model can be written as follows:

$$\begin{cases} \sigma_r^e - \sigma_0 = 2G_\infty \varepsilon_r^e + 2\eta_{ret} G_\infty \dfrac{\partial \varepsilon_r^e}{\partial t} \\ \sigma_\theta^e - \sigma_0 = 2G_\infty \varepsilon_\theta^e + 2\eta_{ret} G_\infty \dfrac{\partial \varepsilon_\theta^e}{\partial t} \end{cases} \tag{8}$$

where $\sigma_r^e$ and $\sigma_\theta^e$ are the radial and tangential stresses in the viscoelastic zone, respectively, and $\eta_{ret}$ is the relaxation time.

If the volume of the viscoelastic zone is unchanged,

$$\varepsilon_r^e + \varepsilon_\theta^e = \frac{du}{dr} + \frac{u}{r} = 0. \tag{9}$$

The displacements at junction $R_p(t)$ between the viscoelastic zone and the plastic softening zone are equal. Thus,

$$\begin{cases} \sigma_r^e = \sigma_0 - \dfrac{MR_p^2(t)}{r^2} \\ \sigma_\theta^e = \sigma_0 + \dfrac{MR_p^2(t)}{r^2}, \\ u^e = \dfrac{A(t)R_p^2(t)}{r} \end{cases} \tag{10}$$

where $A(t)$ is a function of time. In addition,

$$A(t) = \frac{M}{2} \left\{ \frac{1}{G_\infty} \left[ 1 - e^{\left(-\frac{t}{\eta_{ret}}\right)} \right] + \frac{1}{G_0} e^{\left(-\frac{t}{\eta_{ret}}\right)} \right\}, \tag{11}$$

$$M = \frac{\sigma_0(K_p - 1) + \sigma_c}{K_p + 1}. \tag{12}$$

**(2) Plastic softening zone.** Dilatancy occurs in the plastic softening zone. According to formulas (2) and (6), the non-associated flow rule [30], and the equal displacements at the junction between the viscoelastic zone and the plastic softening zone, the displacement equation in the plastic softening zone can be obtained as follows:

$$u^p = \frac{2A(t)r}{\eta_1 + 1} \left( \frac{R_p(t)}{r} \right)^{\eta_1 + 1} + \frac{\eta_1 - 1}{\eta_1 + 1} A(t)r. \tag{13}$$

The strain of the plastic softening zone can be obtained according to formula (13) and the geometric Eq (2):

$$\begin{cases} \varepsilon_r^p = -\dfrac{2A(t)\eta_1}{\eta_1 + 1} \left( \dfrac{R_p(t)}{r} \right)^{\eta_1 + 1} + \dfrac{\eta_1 - 1}{\eta_1 + 1} A(t) \\[3mm] \varepsilon_\theta^p = \dfrac{2A(t)}{\eta_1 + 1} \left( \dfrac{R_p(t)}{r} \right)^{\eta_1 + 1} + \dfrac{\eta_1 - 1}{\eta_1 + 1} A(t) \end{cases}. \tag{14}$$

The following can be obtained according to Eqs (4) and (14):

$$\sigma_c^p = \sigma_c - \frac{2A(t)M_c}{\eta_1 + 1} \left[ \left( \frac{R_p(t)}{r} \right)^{\eta_1 + 1} - 1 \right]. \tag{15}$$

The following can be obtained by solving Eqs (15) and (1) simultaneously, and by making using of the boundary condition $(\sigma_r^e)_{r=R_p(t)} = (\sigma_r^p)_{r=R_p(t)}$.

$$\begin{cases} \sigma_r^p = \dfrac{2}{K_p + 1} \left[ \sigma_0 + \dfrac{\sigma_c}{K_p - 1} + \dfrac{(K_p + 1)M_c A(t)}{(K_p - 1)(K_p + \eta_1)} \right]. \\[3mm] \left( \dfrac{r}{R_p(t)} \right)^{K_p - 1} + \left[ \dfrac{1}{K_p + \eta_1} \left( \dfrac{R_p(t)}{r} \right)^{1 + \eta_1} - \dfrac{1}{K_p - 1} \right]. \\[3mm] \dfrac{2M_c A(t)}{1 + \eta_1} - \dfrac{\sigma_c}{K_p - 1} \\[3mm] \sigma_\theta^p = K_p \sigma_r^p + \sigma_c - \dfrac{2A(t)M_c}{\eta_1 + 1} \left[ \left( \dfrac{R_p(t)}{r} \right)^{\eta_1 + 1} - 1 \right] \end{cases} \tag{16}$$

where $\sigma_r^p$ and $\sigma_\theta^p$ are the radial and tangential stresses of coal in the plastic softening zone, respectively.

**(3) Damaged zone.** The stress expression in the damaged zone can be obtained by solving yield condition (5) in the damaged zone and equilibrium Eq (1) simultaneously, and by

making using of the boundary condition $(\sigma_r^b)_{r=r_0} = 0$:

$$
\begin{cases}
\sigma_r^b = \dfrac{\sigma_c^*}{K_p - 1}\left[\left(\dfrac{r}{r_0}\right)^{K_p-1} - 1\right] \\[4mm]
\sigma_\theta^b = \dfrac{K_p \sigma_c^*}{K_p - 1}\left[\left(\dfrac{r}{r_0}\right)^{K_p-1} - 1\right] + \sigma_c^*
\end{cases}
\tag{17}
$$

Dilatancy occurs in the damaged zone. According to formulas (2) and (6), and the non-associated flow rule and equal displacements at the junction between the plastic softening zone and the damaged zone, the displacement equation of coal in the damaged zone is

$$
u^b = 2A(t)r\left\{\left\{\frac{1}{1+\eta_1} + \frac{1}{1+\eta_2}\left[\left(\frac{R_b(t)}{r}\right)^{1+\eta_2} - 1\right]\right\} \cdot \left(\frac{R_p(t)}{R_b(t)}\right)^{1+\eta_1} + \frac{\eta_1 - 1}{2(1+\eta_1)}\right\},
\tag{18}
$$

where $R_b(t)$ is the radius of the damaged zone.

## 3.2. Calculation model of borehole diameter variation

**(1) Radius of plastic softening zone and damaged zone.** At the junction of the plastic softening zone and the damaged zone, namely, at $r = R_b(t)$, $\sigma_c^p = \sigma_c^*$. According to Eq (15), the following can be obtained:

$$
\sigma_c^* = \sigma_c - \frac{2A(t)M_c}{\eta_1 + 1}\left[\left(\frac{R_p(t)}{R_b(t)}\right)^{\eta_1+1} - 1\right]
\tag{19}
$$

The following,

$$
R_p(t) = R_b(t)\left[1 + \frac{(1+\eta_1)(\sigma_c - \sigma_c^*)}{2A(t)M_c}\right]^{\frac{1}{1+\eta_1}},
\tag{20}
$$

holds, given that

$$
N = \left[1 + \frac{(1+\eta_1)(\sigma_c - \sigma_c^*)}{2A(t)M_c}\right]^{\frac{1}{1+\eta_1}}.
\tag{21}
$$

The radial stresses of the surrounding coal at the junction of the damaged zone and the plastic softening zone are equal, and thus the radius $R_b(t)$ of the damaged zone can be obtained as follows:

$$
R_b(t) = r_0\left\{\left\{\frac{\frac{2}{K_p+1}\left[\sigma_0 + \frac{\sigma_c}{K_p-1} + \frac{(K_p+1)M_cA(t)}{(K_p-1)(K_p+\eta_1)}\right] \cdot N^{1-K_p} +}{\frac{2M_cA(t)}{1+\eta_1}\left[\frac{N^{1+\eta_1}}{K_p+\eta_1} - \frac{1}{K_p-1}\right] - \frac{\sigma_c}{K_p-1}}\right\} \cdot \frac{K_p-1}{\sigma_c^*} + 1\right\}^{\frac{1}{K_p-1}}
\tag{22}
$$

The radius $R_p(t)$ of the plastic softening zone is

$$
R_p(t) = NR_b(t)
\tag{23}
$$

**(2) Calculation model of the borehole diameter over time.** According to Eq (18), the displacement equation of the borehole wall is

$$u_0 = \left\{ \left\{ \frac{1}{1+\eta_1} + \frac{1}{1+\eta_2} \left[ \left( \frac{R_b(t)}{r_0} \right)^{1+\eta_2} - 1 \right] \right\} \left( \frac{R_p(t)}{R_b(t)} \right)^{1+\eta_1} + \frac{\eta_1 - 1}{2(1+\eta_1)} \right\} .2A(t)r_0 \quad (24)$$

The displacement of the borehole wall is a function of time. As times passes, the displacement of the borehole wall increases, and the borehole diameter gradually decreases, even blocking the gas extraction channel. The radius of the borehole at different extraction times can be obtained through formula (24).

$$r_0' = r_0 - u_0 \quad (25)$$

## 4. Example analysis

### 4.1. Numerical simulation parameters

The coal conditions of coal seam $II_1$ of the Ligou Coal Mine are taken as an example. The coal seam is 5.0 m thick. A borehole with a diameter of 94 mm is arranged in the middle of the coal seam with a burial depth of 500 m. The initial geo-stress is 13.5 MPa. After measurement, the Protodyakonov coefficient $f$ is 0.6, the uniaxial compressive strength $\sigma_c$ is 6.15 MPa, the residual strength $\sigma_c^*$ is 0.4 MPa, the internal friction angle $\phi$ is 30.5˚, the initial shear modulus $G_0$ is 1000 MPa, and the long-term shear modulus $G_\infty$ is 500 MPa for a hard coal seam. The $f$ value is 0.3, the uniaxial compressive strength $\sigma_c$ is 3.17 MPa, the residual strength $\sigma_c^*$ is 0.2 MPa, the internal friction angle $\phi$ is 30˚, the initial shear modulus $G_0$ is 800 MPa, and the long-term shear modulus $G_\infty$ is 400 MPa for a soft coal seam.

### 4.2. Comparative analysis of the stress distribution and pressure relief effect of the boreholes in soft and hard coal seams

The mathematical model established above is solved using COMSOL software to obtain the initial stress curve of the surrounding coal in the soft and hard coal seams during the initial stage, as shown in Figs 3 and 4.

As shown in Figs 3 and 4, the radial stress of the surrounding coal in the soft and hard coal seams increases gradually to an initial stress 13.5 MPa, although the growth rate is different. The radial stress in the hard coal seam reaches the initial stress state more quickly. The tangential stress of the surrounding coal in both the soft and hard coal seams gradually increases from the borehole wall to a deeper level, and reaches the maximum at the radius position of the plastic zone. It then gradually reduces to the initial stress of 13.5 MPa. The maximum tangential stress is 20.67 MPa for the coal around the soft coal seam, and 21.82 MPa for that around the hard coal seam. This indicates that a concentration of stress is easily formed in the hard coal seam.

From the radius of the plastic softening zone, the initial radius of the plastic softening zone is 0.405 m in the soft coal seam, and 0.224 m in the hard coal seam. This indicates that the borehole in the soft coal seam had a better pressure relief.

### 4.3. Shrinkage rule of borehole diameter

The displacements of the borehole wall buried at a depth of 500 m for different extraction times can be obtained by solving Eq (24). The results are shown in Fig 5.

According to Fig 5, the surrounding coal in both the soft and hard coal seams may incur creep deformation, leading to a gradual increase in the borehole wall displacement and a

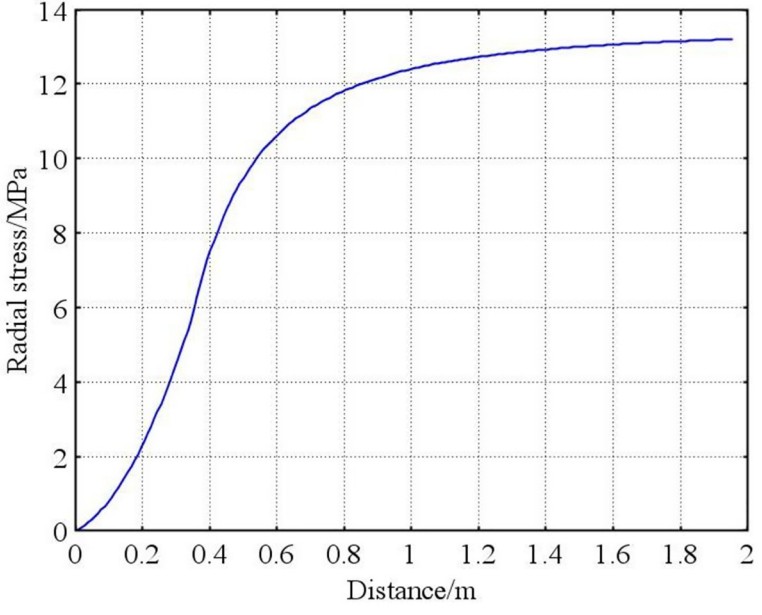

(a) Radial stress

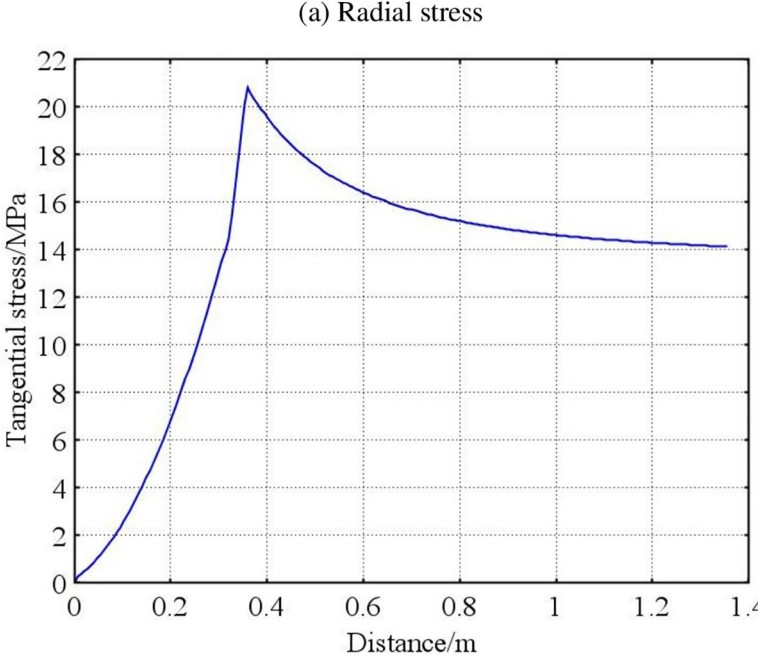

(b) Tangential stress

**Fig 3. Stress distribution of surrounding coal in soft coal seam.** (a) Radial stress and (b) Tangential stress.

gradual reduction in the borehole diameter. However, the shrinkage amplitude of the borehole diameter varies widely in the soft and hard coal seams: For a borehole with an initial radius of 47 mm, the initial displacement in the hard coal seam is 6.58 mm, which then gradually increases; however, the increase in amplitude gradually reduces, becoming stable at 15.2 mm after 60 d, during which the borehole radius shrinks to 31.8 mm. The initial displacement of the borehole in the soft coal seam reaches up to 28.1 mm, then gradually increases, reaching 47

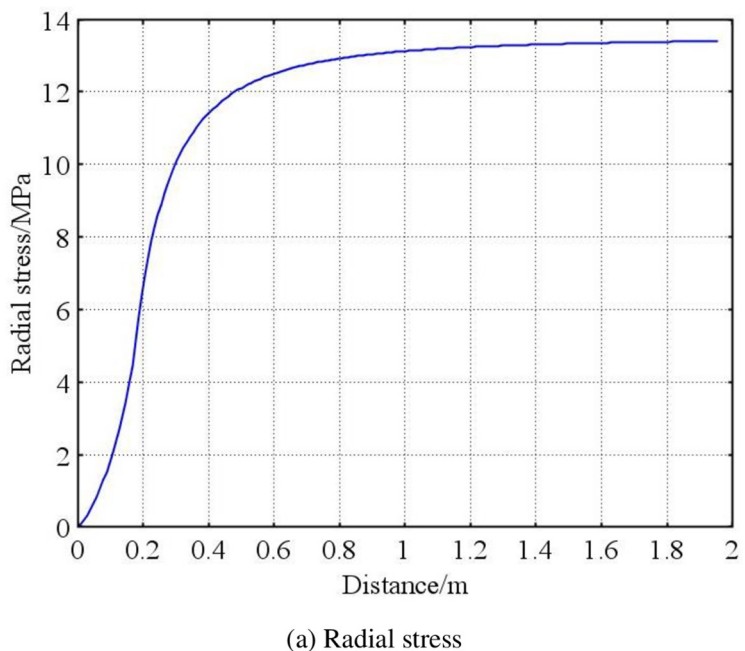

(a) Radial stress

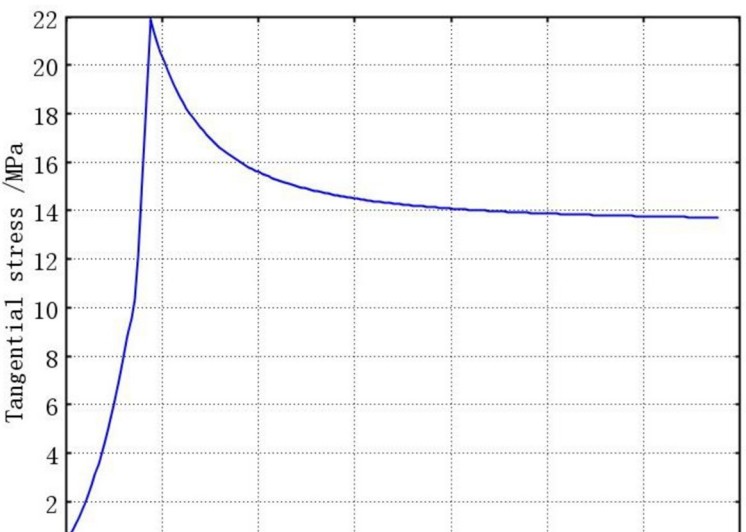

(b) Tangential stress

**Fig 4. Stress distribution of surrounding coal in hard coal seam.** (a) Radial stress and (b) Tangential stress.

mm after only 12 d. The borehole diameter shrinks to zero, which means the borehole is completely blocked. It can also be seen from Fig 5 that the borehole diameters in both the soft and hard coal seams will incur shrinkage over time; however, the hard coal seam shows higher strength and a strong ability to resist damage. Although the borehole diameter shrinks, it remains in a stable state, maintaining a good extraction channel and thus allowing high-concentration gas to be extracted for a long period of time. The surrounding coal in the soft coal

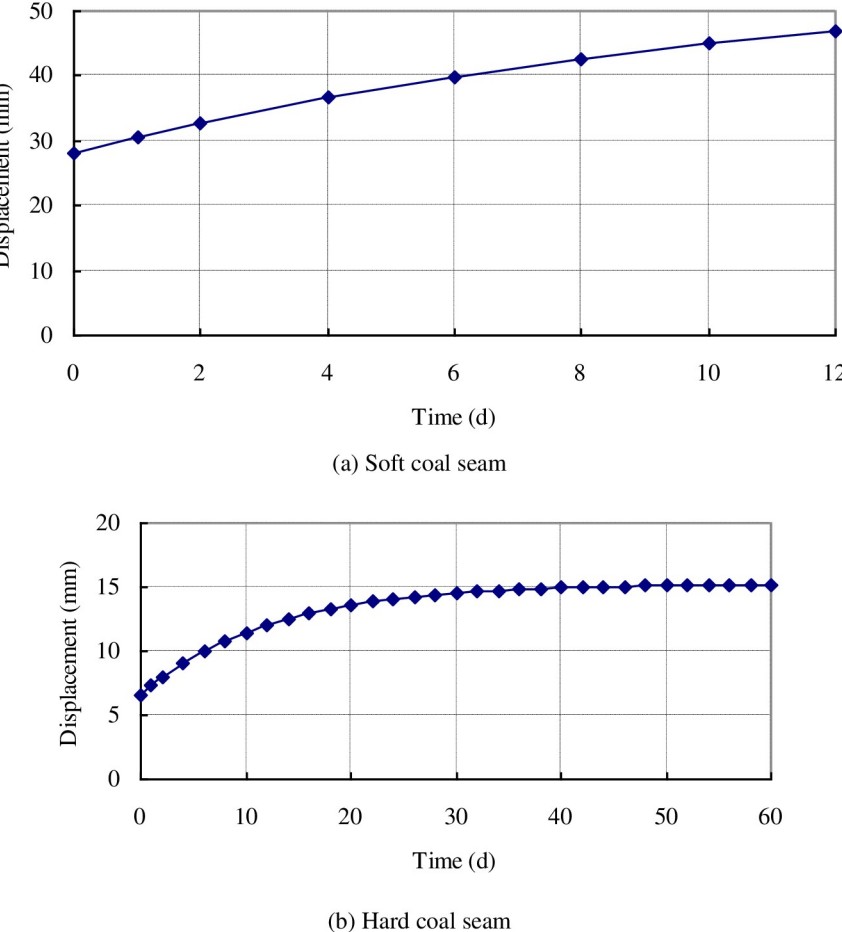

(a) Soft coal seam

(b) Hard coal seam

**Fig 5. Displacements of borehole wall in soft and hard coal seams at different extraction times.** (a) Soft coal seam and (b) Hard coal seam.

seam incurs serious creep deformation, and may experience an instability failure within a short period of time, thus blocking the gas extraction channel and leading to a rapid reduction in the gas extraction concentration.

The gas extraction concentration curves of the soft and hard coal seams in Ligou Mine at different times is shown in Fig 6, which shows that the gas extraction concentration from a borehole in hard coal seams remains at over 40% after 1 months of extraction, whereas that in soft coal seam is reduced to 5% and below after only 15 days. One of the main reasons for this is the serious creep deformation of the borehole in the soft coal seams, which blocks the extraction channel. As a result, it is impossible to effectively extract the gas inside the coal seam. This is consistent with the research results of the present paper.

## 4.4. Influence of burial depth on borehole deformation

With the growth of the burial depth, the initial stress of the surrounding coal gradually increases, and the creep deformation of the coal becomes more intense. Taking the coal conditions of the hard coal seam in the Ligou Coal Mine as an example, when the burial depth is increased from 500 to 800 m, the total stress increases from 13.5 to 21.6 MPa; the change in displacement of a borehole wall with an initial diameter of 97 mm over time is as shown in Fig 7.

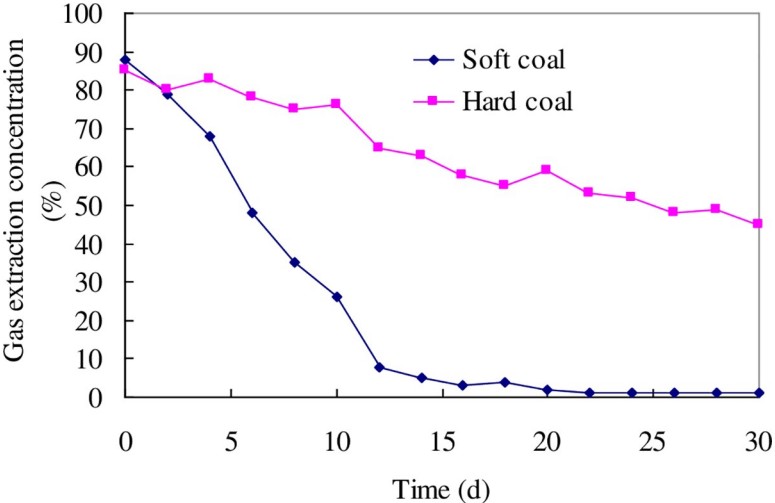

**Fig 6.** The gas extraction concentration curves of the soft and hard coal seams in Ligou Mine with different times.

According to Fig 7, for a hard coal seam with a firmness coefficient of coal $f$ of 0.6, when the burial depth is increased to 800 m, the geo-stress increases to 21.6 MPa, and the initial displacement of the borehole wall increases to 22.99 mm. With an increase in the extraction time, the displacement of the borehole wall reaches 47 mm after only 32 days. At this time, the borehole is completely blocked. Thus, with the increase in burial depth, the creep deformation of the borehole in a hard coal seam will be more intense, leading to blockage. When determining the gas extraction parameters of a deep coal seam, the blockage caused by the rheological characteristics of the coal should also be taken into account for the borehole layout in a hard coal seam.

## 5. Technology of improving the gas extraction effect in soft coal seams

Permeability and drainage channels are the two main factors that affect the gas extraction effect. When a good extraction channel is maintained, the higher the permeability, the better

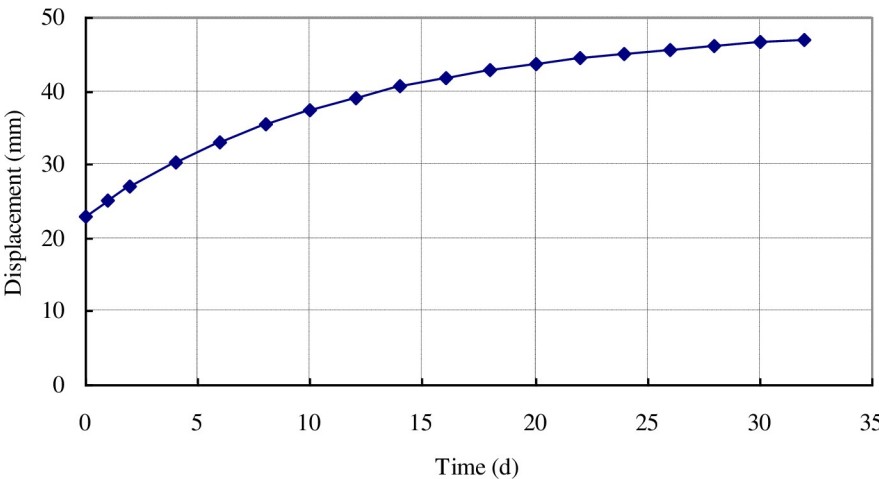

**Fig 7.** Displacement of borehole wall in hard coal seam buried 800 m deep.

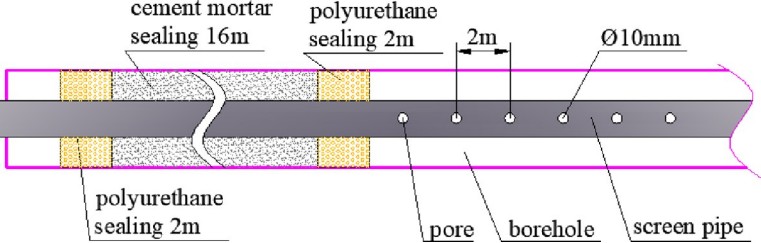

**Fig 8.** Screen pipe installation.

the gas extraction effect; when the extraction channel is blocked, even the permeability is very high, and it is difficult to effectively extract gas. The key to improving the gas extraction effect in soft coal seams is to maintain a smooth extraction channel. Therefore, we have developed a technology, which can effectively improve the gas extraction effect in soft coal seams by installing a screen pipe in the borehole and manually leaving a extraction channel. The screen pipe is made of PVC pipe with a diameter of 35 mm, a pore distance of 2 m and pore diameter of 10 mm (Fig 8). Because it is difficult to install the screen pipes in soft coal seams, we installed screen pipes inside the drill pipe. After the borehole is completed, the drill pipe still remains, and the screen pipe is installed from the inside of the drill pipe and fixed with the aid of hole-bottom suspension device, and then the drill pipe is taken out, and the screen pipe is left in the borehole, thus creating an extraction channel artificially. The borehole is sealed with two plugs and one injection. The two plugging sections are sealed with polyurethane with a sealing length of 2m, and the injection section is 16m long with cement mortar.

The gas extraction scalar with or without screen pipe is shown in Fig 9. It can be seen from the Fig 9 that the average gas extraction increases from 0.043 m³/min to 0.12 m³/min, an increase of 2.77 times.

## 6. Conclusion

1. A visco-elastoplastic model for a borehole considering strain softening and the dilatancy characteristic was established to obtain expressions of the coal stress, diameter variation, and range of pressure relief. The radius of the plastic softening zone was obtained as 0.405

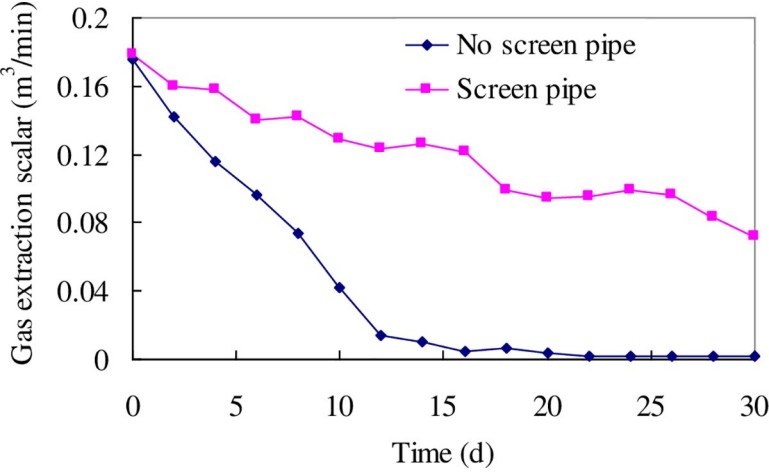

**Fig 9.** Comparison of gas extraction scalar with or without screen pipe.

m for a borehole in a soft coal seam at the test site, and 0.224 m for that in a hard coal seam. This indicates that the borehole in the soft coal seam achieves a better pressure relief.

2. The borehole diameter shrinkage rule over time was determined. The boreholes in both the soft and hard coal seams incurred shrinkage over time. The creep deformation of the borehole in the soft coal seam is more severe, and its diameter shrunk to zero after only 12 days, blocking the gas extraction channel and causing a rapid reduction in the gas extraction concentration. The diameter of the borehole in the hard coal seam also shrunk, but remains in a stable state without blockage, allowing high-concentration gas to be extracted during a long period of time, which is the main reason for the significant difference in the extraction between the soft and hard coal seams.

3. Geo-stress and coal strength are two main factors controlling the amplitude of borehole shrinkage. Owing to an increase in stress, the borehole in a hard coal seam has more intense creep deformation in a deep mine, which may lead to blockage. Thus, the layout of the boreholes should be encrypted in both soft and deep coal seams.

4. The key to improving the gas extraction effect in soft coal seams is to maintain a smooth extraction channel. The screen pipe is installed through the drill pipe to retain an extraction channel, leading to an average gas extraction increase from 0.043 $m^3$/min to 0.12 $m^3$/min, an increase of 2.77 times.

## Author Contributions

**Conceptualization:** Lijuan Sun.

**Data curation:** Fuchang Hao, Lijuan Sun, Fajun Zhao.

**Formal analysis:** Fuchang Hao, Fajun Zhao.

**Project administration:** Fuchang Hao.

**Software:** Lijuan Sun.

**Writing – original draft:** Fuchang Hao.

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
