## [Decision Letter · Decision Letter 0]

15 Jul 2020

PONE-D-20-18356

Borehole diameter shrinkage rule considering rheological properties and its effect on gas extraction

PLOS ONE

Dear Dr. Hao,

Thank you for submitting your manuscript to PLOS ONE. After careful consideration, we feel that it has merit but does not fully meet PLOS ONE’s publication criteria as it currently stands. Therefore, we invite you to submit a revised version of the manuscript that addresses the points raised during the review process.

We look forward to receiving your revised manuscript.

Kind regards,

Jianguo Wang, PhD

Academic Editor

PLOS ONE

Journal Requirements:

Reviewers' comments:

Reviewer's Responses to Questions

**Comments to the Author**

1. Is the manuscript technically sound, and do the data support the conclusions?

Reviewer #1: Yes

Reviewer #2: Yes

2. Has the statistical analysis been performed appropriately and rigorously? 

Reviewer #1: Yes

Reviewer #2: Yes

3. Have the authors made all data underlying the findings in their manuscript fully available?

Reviewer #1: Yes

Reviewer #2: Yes

4. Is the manuscript presented in an intelligible fashion and written in standard English?

Reviewer #1: Yes

Reviewer #2: No

5. Review Comments to the Author

Reviewer #1: 1. Please analyze the effect of permeability on gas extraction effect.

2. It is recommended to re-consolidate the abstract of the manuscript to make it reflect the subject more accurately.

3. It is recommended to modify the format of the manuscript formula to meet the style of this journal.

4. It is recommended to carefully revise the manuscript references.

5. Some special nouns are unified, such as "ground stress",” geo-stress”.

6. How to judge that the low gas concentration of the boreholes is caused by creep deformation but not other factors ?

7.There are many factors that cause low gas extraction concentration, it is recommended to explain in the introduction.

8. There are some small errors in the paper, such as “The displacements at the junction between the the plastic softening zone and the damaged zone are equal”.It is recommended to review the paper carefully.

Reviewer #2: In the manuscript, a visco-elastoplastic model for boreholes considering the strain softening and dilatancy and rheological properties was established, boreholes diameter shrinkage rule in soft and hard coal seams were comparatively analyzed, the reasons for the rapid reduction in the extraction concentration of the borehole in soft coal seams were described, and a technology of improving gas extraction effect of soft coal seams was developed. This topic is interesting and worth of studying, the results of manuscript are innovative. However, the manuscript needs minor revision before acceptance for publication:

1.Both permeability and borehole plugging have a large impact on the gas extraction effect. Please analyze which factor has the greater influence?

2. How to determine when plugging occurs by numerical calculation?

3. There is a small mistake in section 3.3, “For a borehole with an initial diameter of 97 mm” should be “For a borehole with an initial diameter of 94 mm”.

4. There are several spelling errors, please check them carefully and improve it fluently.

6. PLOS authors have the option to publish the peer review history of their article (what does this mean?). If published, this will include your full peer review and any attached files.

Reviewer #1: No

Reviewer #2: No

---

## [Author Response · Author response to Decision Letter 0]

20 Aug 2020

Dear editor and reviewers,

Thank you very much for the valuable comments of the editor and reviewers. We have revised the manuscript and responds to each point raised by the editor and reviewers.

Reviewer #1: 

1. Please analyze the effect of permeability on gas extraction effect.

Permeability is an important indicator that affects the difficulty of gas extraction. When a smooth extraction channel is maintained, the higher the permeability, the better the gas extraction effect; when the extraction channel is blocked, even if the coal seam permeability is high, It is also difficult to extract gas effectively.

2. It is recommended to re-consolidate the abstract of the manuscript to make it reflect the subject more accurately.

We have re-condensed the abstract of the manuscript.

3. It is recommended to modify the format of the manuscript formula to meet the style of this journal.

We revised the formula of the manuscript in accordance with the format requirements of PLOS ONE.

4. It is recommended to carefully revise the manuscript references.

We have revised the manuscript references in accordance with the format requirements of PLOS ONE.

5. Some special nouns are unified, such as "ground stress",” geo-stress”.

We have unified the nouns exclusive to the manuscript.

6. How to judge that the low gas concentration of the boreholes is caused by creep deformation but not other factors ?

Maintaining a smooth extraction channel is the key to improving gas extraction effect. The coal body around the borehole will produce creep deformation, which may block the gas extraction channel. Once the extraction channel is blocked, it is difficult to improve the extraction effect even if the extraction time is extended.

7.There are many factors that cause low gas extraction concentration, it is recommended to explain in the introduction. 

The main factors affecting the concentration of gas extraction are gas extraction channels, coal seam permeability, borehole sealing quality, coal seam gas content and other factors. Various factors have been analyzed in the introduction of the manuscript.

8. There are some small errors in the paper, such as “The displacements at the junction between the the plastic softening zone and the damaged zone are equal”.It is recommended to review the paper carefully.

We carefully revised the content of the paper, found several errors, and showed them in the revised manuscript.

Reviewer #2: 

In the manuscript, a visco-elastoplastic model for boreholes considering the strain softening and dilatancy and rheological properties was established, boreholes diameter shrinkage rule in soft and hard coal seams were comparatively analyzed, the reasons for the rapid reduction in the extraction concentration of the borehole in soft coal seams were described, and a technology of improving gas extraction effect of soft coal seams was developed. This topic is interesting and worth of studying, the results of manuscript are innovative. However, the manuscript needs minor revision before acceptance for publication:

1.Both permeability and borehole plugging have a large impact on the gas extraction effect. Please analyze which factor has the greater influence?

Permeability and extraction channels are the two main factors that affect the effect of gas extraction. When a smooth extraction channel is maintained, the higher the permeability, the better the gas extraction effect; when the extraction channel is blocked, even the permeability is very high and it is difficult to effectively extract gas.

2. How to determine when plugging occurs by numerical calculation?

We have established a viscoelastic-plastic mechanical model of the coal body around the borehole, obtained the expression of the borehole wall displacement, and analyzed the variation of the borehole diameter with time. When the borehole diameter is reduced to 0, the borehole is considered to be blocked.

3. There is a small mistake in section 3.3, “For a borehole with an initial diameter of 97 mm” should be “For a borehole with an initial diameter of 94 mm”.

In response to this error, we revised it in the revised manuscript.

4. There are several spelling errors, please check them carefully and improve it fluently.

We have invited Elsevier to re-edit and polish the manuscript.

Best regards.

Fuchang Hao

---

## [Decision Letter · Decision Letter 1]

28 Aug 2020

Borehole diameter shrinkage rule considering rheological properties and its effect on gas extraction

PONE-D-20-18356R1

Dear Dr. Hao,

We’re pleased to inform you that your manuscript has been judged scientifically suitable for publication and will be formally accepted for publication once it meets all outstanding technical requirements.

Kind regards,

Jianguo Wang, PhD

Academic Editor

PLOS ONE

Additional Editor Comments (optional):

Reviewers' comments:

Reviewer's Responses to Questions

**Comments to the Author**

1. If the authors have adequately addressed your comments raised in a previous round of review and you feel that this manuscript is now acceptable for publication, you may indicate that here to bypass the “Comments to the Author” section, enter your conflict of interest statement in the “Confidential to Editor” section, and submit your "Accept" recommendation.

Reviewer #1: All comments have been addressed

Reviewer #2: All comments have been addressed

2. Is the manuscript technically sound, and do the data support the conclusions?

Reviewer #1: Yes

Reviewer #2: Yes

3. Has the statistical analysis been performed appropriately and rigorously? 

Reviewer #1: Yes

Reviewer #2: Yes

4. Have the authors made all data underlying the findings in their manuscript fully available?

Reviewer #1: Yes

Reviewer #2: Yes

5. Is the manuscript presented in an intelligible fashion and written in standard English?

Reviewer #1: Yes

Reviewer #2: Yes

6. Review Comments to the Author

Reviewer #1: (No Response)

Reviewer #2: (No Response)

7. PLOS authors have the option to publish the peer review history of their article (what does this mean?). If published, this will include your full peer review and any attached files.

Reviewer #1: No

Reviewer #2: No

---

## [Editor Report · Acceptance letter]

15 Sep 2020

PONE-D-20-18356R1 

Borehole diameter shrinkage rule considering rheological properties and its effect on gas extraction 

Dear Dr. Hao:

I'm pleased to inform you that your manuscript has been deemed suitable for publication in PLOS ONE. Congratulations! Your manuscript is now with our production department. 

Kind regards, 

on behalf of

Dr. Jianguo Wang 

Academic Editor

PLOS ONE